# Sweet Potato Is Not Simply an Abundant Food Crop: A Comprehensive Review of Its Phytochemical Constituents, Biological Activities, and the Effects of Processing [note 1]

**DOI:** 10.3390/antiox11091648

**Published:** 2022-08-25

**Authors:** Emily P. Laveriano-Santos, Anallely López-Yerena, Carolina Jaime-Rodríguez, Johana González-Coria, Rosa M. Lamuela-Raventós, Anna Vallverdú-Queralt, Joan Romanyà, Maria Pérez

**Affiliations:** 1Department of Nutrition, Food Science and Gastronomy XIA, Faculty of Pharmacy and Food Sciences, University of Barcelona, 08028 Barcelona, Spain; 2Institute of Nutrition and Food Safety (INSA-UB), University of Barcelona, 08028 Barcelona, Spain; 3CIBER Physiopathology of Obesity and Nutrition (CIBEROBN), Institute of Health Carlos III, 28220 Madrid, Spain; 4Department of Biology, Health, and Environment, Faculty of Pharmacy and Food Sciences, Institute of Nutrition and Food Safety (INSA-UB), University of Barcelona, 08028 Barcelona, Spain

**Keywords:** *Ipomoea batata* L. roots, bioactive compounds, biological functions, polyphenols, healthy food, carotenes, cooking

## Abstract

Nowadays, sweet potato (*Ipomoea batata* L.; Lam.) is considered a very interesting nutritive food because it is rich in complex carbohydrates, but as a tubercle, contains high amounts of health-promoting secondary metabolites. The aim of this review is to summarize the most recently published information on this root vegetable, focusing on its bioactive phytochemical constituents, potential effects on health, and the impact of processing technologies. Sweet potato is considered an excellent source of dietary carotenoids, and polysaccharides, whose health benefits include antioxidant, anti-inflammatory and hepatoprotective activity, cardiovascular protection, anticancer properties and improvement in neurological and memory capacity, metabolic disorders, and intestinal barrier function. Moreover, the purple sweet potato, due to its high anthocyanin content, represents a unique food option for consumers, as well as a potential source of functional ingredients for healthy food products. In this context, the effects of commercial processing and domestic cooking techniques on sweet potato bioactive compounds require further study to understand how to minimize their loss.

## 1. Introduction

Sweet potato (*Ipomoea batata* L.; Lam.) is a dicotyledonous vegetable belonging to the family Convolvulaceae. It is the seventh most produced crop worldwide after wheat, rice, maize, potato, barley, and cassava, and the fifth in developing countries [1]. Sweet potato tubers, leaves, and shoots are good sources of nutrients for humans and animals, with around 50% of the crop used for animal feed.

Sweet potato tubers contain macronutrients such as starch, dietary fiber, and protein, in addition to an extensive range of micronutrients, including minerals (manganese, copper, potassium, and iron), vitamins (mainly B complex, C, and E), and provitamin A (as carotenoids), anthocyanins (purple sweet potatoes), flavonoids, and coumarins [2]. Compared to other root and tuber crops, the sweet potato contains more carbohydrates and proteins, as well as certain vitamins and minerals [3], and it has higher levels of provitamin A, vitamin C, and minerals than wheat or rice [4].

Due to its content of several bioactive secondary metabolites, the sweet potato is attracting the attention of the food industry, consumers, and scientists [5], not only as a healthy product but also as an ingredient for functional foods [6]. These phytochemicals provide physiological benefits that ultimately, either individually or collectively, promote health and longevity in consumers [7].

The color of this food is linked to its beneficial health effects [6]. Lighter fleshed varieties are reported to have higher levels of phenolic compounds, whereas a more intense yellow color is associated with a higher content of carotenoids, mainly ß-carotene [8]. Additionally, yellow- and orange-fleshed sweet potatoes are rich in phenolic acids, while those varieties that are purple have very high levels of anthocyanins [9,10].

Recent reviews on sweet potato leaves covered current knowledge of their bioactive composition and possible health effects [11] and the protocols developed for plant regeneration as an alternative method to produce disease-free planting material [12]. Beyond the widely studied potential of sweet potato leaves and their cultivars practices, sweet potato roots are more than just foods used for centuries as a major source of carbohydrates. Today they are recognized as a highly nutritious and useful food for the prevention of chronic diseases. To corroborate this, we have integrated for the first time updated information on their content in bioactive compounds, and an extensive revision of the in vitro and in vivo evidence of its benefits for human health. In addition, since phenolic compounds are important constituents of a healthy human diet, we have explored the effect of sweet potato processing on phenolic compounds.

## 2. Bioactive Compounds in Sweet Potato Tubers

Sweet potatoes are a good source of several bioactive compounds, above all (poly)phenols, terpenoids, tannins, saponins, glycosides, alkaloids, and phytosterols. The diversity of skin and flesh color in this root vegetable arises from the different levels of (poly)phenols and carotenoids [9,13,14,15,16,17]. Thus, the dominant pigments in purple sweet potatoes are anthocyanins and phenolic acids, whereas, in yellow and orange-fleshed sweet potatoes, they are phenolic acids, flavonoids, and carotenoids [13,14]. In addition to genetics, the concentration and bioavailability of bioactive compounds in sweet potatoes and derived products are affected by external factors such as environment and cultivar, storage conditions, and processing; moreover, the available data can be influenced by the extraction and analytical methods [18,19]. 

### 2.1. (Poly)phenols

Sweet potatoes are characterized by a high concentration of flavonoids and phenolic acids [14]. Flavonoids are mainly found in purple-fleshed potatoes in the form of anthocyanins and quercetin glycosides [13,14]. The yellow- and orange-fleshed tubers contain a mixture of phenolic acids, above all caffeic acid, chlorogenic acid, and caffeoylquinic acid derivatives [14].

#### 2.1.1. Total (Poly)phenol Content 

The total (poly)phenol content (TPC) in sweet potato flesh has been determined by several authors using a simple spectrophotometric method, and the reported values are present in a broad range from 10 to 408 mg of chlorogenic acid equivalents/100 g of fresh weight [9,20,21] or from 1.8 to 136.1 mg of gallic acid equivalents/100 g fresh weight (Table 1, Table 2 and Table 3) [13,22]. The wide variability of (poly)phenol content in sweet potatoes is associated with their genetic diversity, being higher in those with light orange, white, and yellow flesh [9]. This variation can also be influenced by environmental factors such as the type of soil, sun exposure, rainfall, and level of ripeness, as well as the cultivation method [13,23]. (Poly)phenol concentrations in sweet potatoes are lower in locations with more daylight hours and cooler temperatures. Thus, the TPC of commercial sweet potatoes cultivated in the US ranged from 57.1 to 78.6 mg of chlorogenic acid equivalents/100 g of fresh weight [20], whereas orange sweet potatoes grown in Bangladesh contained about 94.3–136.1 mg gallic acid equivalents/100 g fresh weight (Table 1) [13]. Moreover, food processing techniques, including cooking, can also alter the TPC in this root vegetable [1,18,24].

**Table 1 antioxidants-11-01648-t001:** (Poly)phenol content of orange sweet potato flesh.

Origin	Sample Extraction	Analytical Method	Phytochemical	Amount of Phytochemical	Ref.
USA	Methanol (80%)	Folin–Ciocalteu	TPC	~1.4 mg CA/g FW	[9]
	Methanol (80%)	pH-differential	TAC	<0.1 mg anthocyanins/g FW	
USA	Ethanol (80%)	Folin–Ciocalteu	TPC	~0.1 mg CA/g FW	[20]
Bangladesh	Acetone:water (7:3, *v*/*v*)	Folin-Ciocalteu	TPC	94.3 to 136.1 mg GA/100 g FW	[13]
Kenya	Methanol (80%)	Folin–Ciocalteu	TPC	TPC mg GA/100 g DW: - Raw: 4025.4 to 4999.4 - Boiling: 1222.8 to 4699.7 - Dehydration: 659.3 to 5861.7 - Fermentation and dehydration: 1662.4 to 3962.3	[18,25]
		Aluminum chloride	TPC	TPC mg catechin/100 g DW: - Raw: 1294.3 to 4211.2 - Boiling: 998.2 to 1554.4, - Dehydration: 411.0 to 3816.2 - Fermentation and dehydration: 913.8 to 3270.7	
USA	Ethanol (80%)	Folin–Ciocalteu	TPC	57.1 to 78.6 mg CA/100 g FW	[20]
		HPLC-DAD and LC-MS/MS	Phenolic acids: chlorogenic acid, caffeic acid, di-O-caffeoylquinic acids	Phenolic acids mg/100 g FW: - Chlorogenic acid 5.1 to 9.3 - Caffeic acid 5.1 to 9.3 - 4,5-di-O-caffeoylquinic acid 0.6 to 2.4 - 3,5-di-O-caffeoylquinic acid 2.3 to 6.5 - 3,4-di-O-caffeoylquinic acid 0.2 to 0.5	
Pakistan	Ethyl acetate	Folin–Ciocalteu	TPC	319.8 μg GA/mg DE	[26]
	Methanol			262.6 μg GA/mg DE	
	Ethyl acetate	Aluminum chloride	TFC	208.8 μg quercetin/mg DE	
	Methanol			177.8 μg quercetin/mg DE	
Korea	Methanol (50%) withn1.2 M HCl at 80 °C	HPLC system	Quercetin, myricetin, kaempferol, luteolin, ferulic, *p*-coumaric, *p*-hydroxybenzoic, sinapic, syringic, and vanillic acids.	Flavonoids: 127.1 µg/g DW (Quercetin: 59.9, myricetin: 39.8, kaempferol: 18.9, luteolin: 8.5) Phenolic acids: 71.1 µg/g DW (*p*-hydroxybenzoic acid:7.8, vanillic acid 7.9, syringic acid:3.8, *p*-coumaric:11.9, ferulic acid:24.6, sinapic acid:15.2)	[27]

CA: chlorogenic acid; DE: dry extract; DW: dry weight; FW: Fresh weight; GA: gallic acid, HPLC: high-performance liquid chromatography; MS: mass spectrometry; TPC: Total phenolic content; TFC: Total flavonoid content; TAC: Total anthocyanin content.

#### 2.1.2. Flavonoids 

Quercetin, myricetin, luteolin, kaempferol, and apigenin are the flavonoids identified in sweet potatoes, especially in varieties with orange and purple flesh (Figure 1). As mentioned, the flavonoid content depends on the variety, being higher in purple-fleshed (579.5 µg/g dry weight) than orange- (121.1 µg/g dry weight) and white-fleshed (45.4 µg/g dry weight) sweet potatoes [27]. The major flavonoid in purple and orange potatoes is quercetin, followed by myricetin, kaempferol, and luteolin [27]. 

#### 2.1.3. Anthocyanins (In Purple Sweet Potato)

Anthocyanins are the flavonoid family responsible for the purple coloration of sweet potato flesh and skin. Total anthocyanin concentrations are higher in the purple varieties than those with orange flesh, with reported values of 14–182 mg/100 g fresh weight (Table 2) [9,21,31,38,39]. More than 20 anthocyanins have been identified in sweet potatoes [17,33,34,35,40,41,42]. The main ones in purple varieties are 3-sophoroside-5-glucoside derivatives of peonidin, cyanidin, and pelargonidin aglycones (Figure 2), almost all of them mono- or di-acylated with *p*-hydroxybenzoic acid, ferulic acid, or caffeic acid [16,29,32,33,34,37,40,42]. According to Fossen et al., these acylated forms represent more than 98% of the total anthocyanin content in purple sweet potato [43]. However, the anthocyanin composition differs widely between varieties. Thus, Im et al. reported that ‘Sinjami’, ‘Danjami’, and ‘Yeonjami’ contained 72–77% di-acylated anthocyanins compared to 90–95% in the Korean varieties ‘Jami’ and ‘Borami’ [22]. Conversely, a higher proportion of mono-acylated anthocyanins (21–24%) was found in ‘Sinjami’, ‘Danjami’, and ‘Yeonjami’ compared to ‘Jami’ and ‘Borami’ [22]. Acylation with the phenolic acids *p*-coumaric, ferulic, or caffeic acids enhances the stability of anthocyanins in conditions of heat, pH, and ultraviolet radiation, which facilitates their application in the food industry as natural colorants [42,44,45]. Compared to non- and di-acylated anthocyanins, mono-acylated forms have a higher resistance to heat, especially those derived from cyanidin 3- -hydroxybenzoylsophoroside-5-glucoside [35]. Therefore, these acylated forms could contribute to the higher antioxidant activity of purple sweet potatoes compared to those of other colours [35]. 

#### 2.1.4. Phenolic Acids

The sensory qualities of sweet potato are associated with their content of phenolic acids, namely chlorogenic, dicaffeoylquinic, caffeic, ferulic, *p*-hydroxybenzoic, coumaric, sinapic, syringic, and vanillic acids (Figure 3) [9,13,20,21,22,26,27,30]. Park et al. reported that the level of these phenolic acids varies with the flesh color, being almost ten-fold higher in purple (744.3 µg/g dry weight) than in orange and white sweet potatoes (71.1 µg/g dry weight and 52.5 µg/g dry weight, respectively) (Table 1, Table 2 and Table 3) [27]. The predominance of chlorogenic, ferulic, coumaric, and caffeic acids in purple-fleshed sweet potatoes is due to the stability afforded by their interaction with anthocyanins [22,27,44,45,46]. 

#### 2.1.5. Isolation, Identification, and Quantification of Phenolic Compounds in Sweet Potatoes

The isolation of total monomeric anthocyanins and other phenolic compounds from sweet potatoes is usually carried out using solvent extraction processes. Methanol and ethanol (80:20, solvent: water) are the most used polar solvents, as they provide higher phenolic extraction efficiency [9,20,30,31,46]. According to the results of Vishnu et al. (2019), the extraction of anthocyanins from purple sweet potato is more efficient when using acidified methanol with 0.5% trifluoroacetic acid compared to ethanol, methanol/trifluoroacetic acid/water, and ethanol/trifluoroacetic acid/water [32]. 

Spectrophotometric methods are used to quantify total (poly)phenols and anthocyanins. Folin–Ciocalteu reagent has been applied to quantify total (poly)phenols in sweet potatoes, commonly expressed as equivalents of chlorogenic acid or gallic acid [13,20,21,22,24,26,30]. Nevertheless, this reagent could provide overestimated results because of a tendency to react with other compounds in sweet potatoes, such as sugars and ascorbic acid [47,48]. Padda et al. reported that the average TPC in sweet potato tubers measured with Folin–Denis and Folin–Ciocalteu reagents were 60.9 and 74.6 mg/100 g fresh weight, respectively, indicating that the Folin–Denis reagent is an alternative option for (poly)phenol quantification in sweet potatoes [46]. Regarding anthocyanins, the total content is commonly quantified using a pH differential-spectrophotometry method reading at ranges from 420 to 730 nm [9,21,30,37]. The major individual phenolic compounds in sweet potatoes are identified and quantified using phase-reverse phase high-performance liquid chromatography (HPLC), electrospray ionization mass spectrometry, and tandem MS [20,22,27,31,32,33,35,40,41,42]. 

### 2.2. Carotenoids

Sweet potato tubers are an excellent source of carotenoids, which contribute to their yellow and orange color (Figure 4). The total carotenoid content differs significantly according to the potato color and variety (Table 4 and Table 5). *β*-carotene is the most abundant in those with orange flesh (more than 99% of total carotenoid content), whereas *trans-β-carotene* predominates in those with white and purple flesh. In orange varieties, Vimala et al. (2011) reported 5.9–12.9 mg/100 g of *β*-carotene [49], and Alam et al. (2016), 0.38–7.38 mg/100 g [13]. Lutein and zeaxanthin have been identified as minor carotenoids in sweet potatoes, with reported ranges of 0.1–0.4 and 0.1–0.2 mg/100 g dry weight, respectively [19,27]. According to the results of Park et al. (2016), lutein and zeaxanthin levels are higher in potatoes with purple versus white flesh [27].

### 2.3. Other Phytochemicals

Orange-fleshed sweet potatoes also contain phytosterols (Figure 5). Daucosterol has been reported at concentrations ranging from 0.01 to 0.5 mg/g dry weight [50], whereas *β*-sitosterol, the most abundant phytosterol in sweet potato (55 to 78% of total phytosterols), is found at ranges of 122–358 mg/kg, followed by campesterol (35–101 mg/kg), stigmasterol (5–58 mg/kg), and isofucosterol (5–23 mg/kg) (Table 4) [51].

**Table 4 antioxidants-11-01648-t004:** Carotenoid, phytosterol, and other phytochemical contents of orange sweet potato flesh.

Origin	Sample Extraction	Analythical Method	Phytochemical	Amount of Phytochemical	Ref.
** *Carotenoids* **					
USA	Hexane-acetone (1:1)	Reverse-phase HPLC	β-Carotene	~122.0 µg/g FW, ~18.2 µg/g FW (light-orange)	[9]
Bangladesh	Hexane-acetone (1:1)	Spectrophotometry	TC	0.38 to 7.24 µ *	[13]
		Reverse-phase HPLC	TC	19.3 to 61.9 µ *	
			*trans*-β-Carotene	76.6 to 96.5 µ *	
			*cis*-β-Carotene:	3.5 to 23.4 µ *	[52]
Brazil	Acetone-petroleum ether. Petroleum ether: diethyl ether (1:1)	Reverse-phase HPLC	*trans-*β-Carotene	Raw: 79.1 to 128.5 mg *, boiled: 68.9 to 133.3 mg *, roasted: 64.6 to 127.0 mg *, steamed: 69.4 to 131.0 mg *, flour: 45.4 to 79.7 mg *	[19]
			*13-cis-*β-Carotene	Raw: 9.3 to 9.6 mg *, boiled: 4.3 to 8.6 mg *, roasted: 4.3 to 11.1 mg *, steamed: 7.1 to 8.5 mg *, flour: 2.7 to 4.7 mg *	
			*9-cis-*β-Carotene	Raw: 4.9 to 6.1 mg *, boiled: 3.9 to 6.0 mg *, roasted: 3.8 to 5.5 mg *, steamed: 5.2 to 7.4 mg *, flour: 1.5 to 2.1 mg *	
			5,6-Eepoxy*-*β-carotene	Raw: 7.0 to 11.3 mg *, boiled: 7.8 to 13.1 mg *, roasted: 7.0 to 9.6 mg *, steamed: 7.6 to 15.4 mg *, flour: 3.8 to 6.5 mg *	
			Lutein	Raw: 0.1 to 0.4 mg *, boiled: 0.2 to 0.4 mg *, roasted: 0.1 to 0.6 mg *, steamed: 0.1 to 1.1 mg *, flour: 0.1 to 0.3 mg *	
			Zeaxanthin	Raw: 0.1 to 0.2 mg *, boiled: 0.1 to 0.3 mg *, roasted: 0.1 to 0.2 mg *, steamed: 0.1 to 0.6 mg *, flour: 0.1 to 0.2 mg *	
Kenya	Methanol and tetrahydrofuran	Reverse-phase HPLC	Lutein	0.01 to 0.1 mg *	[18,25]
			Zeaxanthin	0.02 to 0.5 mg *	
			*β*-Xanthin	0.1 to 0.5 mg *	
			13-*cis*-*β*-Carotene	0.05 to 0.4 mg *	
			All *trans β*-Carotene	2.6 to 18.2 mg *	
			*β*-9-*cis*-*β*-Carotene	0.05 to 0.4 mg *	
Korea	Ethanol (0.1% ascorbic acid)	HPLC system	TC	93.4 µg **	
			Lutein	0.15 µg **	[27]
			α-Carotene	0.44 µg **	
			(all *E*)-β-Carotene	68.74 µg **	
			(9*Z*)-β-Carotene	1.45 µg **	
			(13*Z*)-β-Carotene	22.64 µg **	
India	Hexane-acetone (6:4)	HPLC	TC	7.47 to 15.47 mg/100 g FW	
			β-Carotene	5.85 to 13.63 mg/100 g FW	[46]
** *Phytosterols* **					
China	Ethanol (70%)	HPLC system	Daucosterol linolenate	0.05 to 0.2 mg **	[50]
	Acetone-petroleum ether, ethyl acetate, and n-butanol (1:1)		Daucosterol linoleate	0.2 to 0.5 mg **	
			Daucosterol palmitate	0.3 to 0.6 mg **	
** *Other phytochemicals* **					
Kenya	Water	UV spectrophotometry	Tannic acid	0.04 to 0.13 g *	[25]
	Water	HPLC	Oxalic acid	0.003 to 0.132 g *	
	NS	ELISA kit	Phytic acid	0.05 to 0.42 g *	

HPLC: high-performance liquid chromatography; DW: dry weight; FW: Fresh weight; TC: total carotenoids; * Results expressed per 100 g DW, ** Results expressed per 1 g DW.

**Table 5 antioxidants-11-01648-t005:** Carotenoid content of purple, yellow, and white sweet potato flesh.

Color of Sweet Potato Flesh (Origin)	Sample Extraction	Analythical Method	Phytochemical	Amount of Phytochemical	Ref.
Purple (USA)	Hexane–acetone (1:1)	Reverse-phase HPLC	β-Carotene	~22.5 µg/g FW, ~50.6 µg/g FW (light purple)	[9]
Purple (Korea)	Ethanol (0.1% ascorbic acid)	HPLC system	TC	−2.22 µg **	[27]
			Lutein	−0.28 µg **	
			Zeaxanthin	−0.11 µg **	
			(all *E*)-β-Carotene	−1.53 µg **	
			(9*Z*)-β-Carotene	−0.02 µg **	
			(13*Z*)-β-Carotene	−0.28 µg **	
Yellow (USA)	Hexane–acetone (1:1)	Reverse-phase HPLC	β-Carotene	~−1.9 µg/g FW	[9]
Yellowish cream (Bangladesh)	Acetone:petroleum ether	Spectrophotometry Reverse pase HPLC	TC	−3.3 to 5.6 µ *	[52]
			*trans*--β-Carotene	−83.6 to 84.3 µ *	
			*cis*--β-Carotene:	−13.4 to 15.7 µ *	
White (USA)	Hexane–acetone (1:1)	Reverse-phase HPLC	β-Carotene	−0.2 µg/g FW	[9]
White (Bangladesh)	Acetone:petroleum ether	Spectrophotometry Reverse pase HPLC	TC	−1.0 µ *	[52]
White (Korea)	Ethanol (0.1% ascorbic acid)	HPLC system	TC	−1.37 µg **	[27]
			Lutein	−0.27 µg **	
			Zeaxanthin	−0.03 µg **	
			α-Carotene	−0.01 µg **	
			(all *E*)-β-Carotene	−0.83 µg **	
			(9*Z*)-β-Carotene	−0.09 µg **	
			(13*Z*)-β-Carotene	−0.14 µg **	

HPLC: high-performance liquid chromatography; DW: dry weight; FW: Fresh weight; TC: total carotenoids; * Results expressed per 100 g DW, ** Results expressed per 1 g DW.

## 3. Beneficial Health Effects of Sweet Potatoes 

Traditionally, sweet potatoes have been used as an important source of carbohydrates and energy for both human beings and livestock because of their high content of starch [53]. Nowadays, sweet potatoes are recognized as a highly nutritious and useful food for the prevention of chronic diseases, mainly due to its content of dietary fiber, naturally occurring sugars, protein content, vitamins A and C, potassium, iron, and calcium, and its low amount of fat (mainly saturated fat), sodium, and cholesterol [54].

The regular consumption of sweet potatoes or extracts rich in their bioactive phytochemicals seems to also be beneficial for human health as evidenced by The Okinawan Diet [54]. However, most of the positive health effects reported until now (Figure 6), which will be explained in detail in the following sections, are based on in vitro studies and just a few of them in animal models. So far only one human trial has been carried out.

### 3.1. Antioxidant Properties

The potent antioxidant capacity of purple sweet potatoes is mainly attributed to their anthocyanin and carotenoid content. These compounds exhibit free radical scavenging activity, and consequently, the consumption of anthocyanin-rich products is associated with a lower risk of diabetes cardiovascular disease, cancer, and cognitive performance [55]. Anthocyanins from purple sweet potato showed stronger 1,1-diphenyl-2-picrylhydrazyl (DPPH) radical scavenging capacity than those from red cabbage, grape skin, elderberry, or purple corn, and some were also more active than either vitamin C or vitamin E [56,57]. 

The overall antioxidant capacity of white-fleshed cultivars was attributed to their high content of phenolic acids and carotenoids, and it was concluded that their consumption might protect the human body from oxidative stress [46]. Rather than anthocyanins, chlorogenic acid was the main DPPH radical scavenger in extracts of purple-fleshed varieties “Miyanou-36” and “Bise” [28]. On the other hand, in yellow or orange sweet potatoes, the carotenoid β-carotene, which is the main pigment, has a strong antioxidant capacity due to its conjugated double bonds [58] and acts as a source of provitamin A [8]. 

### 3.2. Hepatoprotective Effects

To date, most of the studies on hepatoprotective effects have been carried out using animal models and only one study has been carried out in human studies. In addition, it should be noted that all these studies have focused on evaluating the protective effect of anthocyanins from sweet potatoes.

The anthocyanins offer protection from injury induced by hepatotoxins mainly by inhibition of lipid peroxidation and scavenging free radicals. In healthy men with borderline hepatitis, purple sweet potato beverages significantly decreased the serum levels of some liver enzymes, particularly gamma-glutamyl transferase (GGT) [59]. 

In animal models, a protective effect against CCl_4_-induced acute liver damage has been extensively reported after the intake of anthocyanin-rich purple sweet potato extract [57,60,61], as well as sweet potato polysaccharides [62]. Inhibition of lipid peroxidation was also observed in male rats fed a high-cholesterol diet after the administration of anthocyanin-rich sweet potato flakes [63] and in the acetaminophen-induced hepatotoxicity mouse model [64]. Purple sweet potato anthocyanin (PSPA) attenuated the oxidative stress and inflammatory response induced by D-galactose in mouse liver [65], whereas an extract from Shinzami, a variety of purple sweet potato, prevented ischaemia–reperfusion-induced liver damage in rats by improvement of the antioxidant status [66]. 

The fact that the protective effect has been demonstrated in animal studies indicates that human studies are needed to show whether there is strong evidence for the hepatoprotective effects of anthocyanins in purple sweet potato.

### 3.3. Cognitive and Memory Improvement 

The bioactive compounds from purple sweet potato exhibit memory-enhancing effects. According to Isoda et al. (2010, 2013), caffeoylquinic acid-rich purple sweet potato extracts improved spatial learning and memory in a mouse model of aging [67,68]. In addition, D-galactose-induced impairment of memory and spatial learning was repaired through the regulation of synaptic protein expression in the hippocampus and cerebral cortex of mice by different PSPA treatments [69]. Protection against Aβ-induced neurotoxicity by caffeoylquinic acids has also been reported in a mouse model [70,71,72]. Similarly, in mice injected with Aβ₁₋₄₂, attenuation of cognitive dysfunction and neuronal cell damage was observed after the administration of 2,4-di-*tert*-butylphenol extracts from sweet potatoes [73]. 

A neuroprotective effect of purple sweet potato anthocyanins in a Wistar rat model with ischemic stroke was reported by Adnyana et al. (2018, 2019), which was attributed to the inhibition of damaging effects of reactive oxygen species (ROS) [74]. The treated rats showed an enhanced neurological score between day-3 and day-7 post-stroke, an increase in the brain-derived neurotrophic factor level, and a reduced apoptosis rate [75].

Cognitive deterioration is also associated with obesity, a growing public health concern. In this context, purple sweet potato anthocyanins were observed to improve cognitive function in high-fat diet-fed mice via the activation of AMP-activated protein kinase, which protects against hippocampal apoptosis by restoring impaired autophagy [76]. The protective role of this pigment in high-fat diet-associated neuroinflammation in the mouse brain was also examined by Li (2018), who found significant improvement in impaired memory function and behavior, as well as suppression of the increment in body weight, hyperlipemia, fat content, and endotoxin levels [77].

In summary, numerous studies have been carried out attempting to explain the neuroprotective effect of sweet potato. However, all studies have only been conducted in animals, which leaves the need to confirm this protective effect in human studies. In addition, future studies are needed to evaluate the brain-protective effect of other sweet potato varieties (yellow- and orange-fleshed sweet potatoes).

### 3.4. In Vitro and In-Vivo Cancer Chemoprevention Capacity 

Most of the studies on protective cancer effects derived from sweet potato consumption have been carried out using in-vitro and animal models. Further research is needed in this field, especially in randomized clinical trials in humans.

Extracts of purple-fleshed sweet potato Tainung 73 (PFSP TNG 73) are reported to have in vitro anti-inflammatory and anticancer activities, attributed to a high content of antioxidative compounds, including phenolics, flavonoids, and the pigment anthocyanin [78]. Anthocyanin-rich extracts of PFSP TNG 73 suppressed the production of nitric oxide and some proinflammatory cytokines, such as NF𝜅-𝛽, TNF-𝛼, and IL-6, in macrophage cells stimulated by lipopolysaccharides. They also inhibited the growth of some in vitro cancer cell lines, including human breast cancer (MCF-7), gastric cancer (SNU-1), and colon adenocarcinoma (WiDr), in a concentration- and time-dependent manner. Additionally, PFSP TNG 73 extracts induced apoptosis in MFC-7 cells. Thus, this variety of sweet potato is a source of metabolites with potential application in the development of drugs, nutritional foods, and health supplements. 

Purple sweet potatoes also contain polysaccharides with promising antitumor properties. Three beta-type polysaccharides, PSPP1-1, PSPP2-1, and PSPP3, with low amounts of proteins and uronic acids, were isolated from crude purple sweet potato polysaccharides. In an in vitro antitumor assay, PSPP1-1 exhibited strong activities against gastric cancer SGC7901 cells and colon cancer SW620 cells, whereas PSPP2-1 and PSPP3-1 had moderate activities. Furthermore, PSPP1-1 was found to induce apoptosis in both types of cancer cells [6]. 

#### 3.4.1. Breast Cancer

Breast cancer is a prominent cause of mortality in women throughout the world. In a study by Kato et al. [79] using E0771 murine breast cancer cells, lipid-soluble polyphenols (mainly caffeic acid derivatives) from fermented sweet potato were found to accumulate in cell cytoplasm due to their high lipophilicity and reduce ROS through their strong antioxidant activity. These metabolites also arrested the cell cycle at G0/G1 by suppressing Akt activity and enhancing the cytotoxicity of anti-cancer agents. Thus, lipid-soluble polyphenols from sweet potatoes inhibited tumor growth and improved the efficacy of chemotherapy drugs, suggesting they have application as a functional food to support cancer therapy. 

Another study revealed that three phytosterols from sweet potato, daucosterol linolenate (DLA), daucosterol linoleate (DL), and daucosterol palmitate (DP), had a stronger inhibitory effect against the MCF-7 than the MDA-MB-231 breast cancer cell line, and had no impact on non-tumorigenic MCF-10A cells [50]. In vivo experiments demonstrated that DLA, DL, and DP suppressed MCF-7 xenografts in nude mice. In another study, sitosterol-d-glucoside (β-SDG), a recently isolated phytosterol from sweet potato, also displayed potent anticancer activity [80] against MCF7 and MDA-MB-231 cell lines and suppressed the growth of MCF7 xenografts in nude mice. This effect of β-SDG was due to the up-regulation of miR-10a expression and inactivation of the PI3K–Akt signaling pathway.

A component of the new sweet potato variety Zhongshu NO. 1, the glycoprotein SPG-56, was reported to inhibit proliferation and promote apoptosis of MCF-7 cells in mice in a dose- and time-dependent manner [81]. The serum tumor markers CEA, CA125, and CA153 were reduced by 54.8%, 91.8%, and 90.3%, respectively, in mice orally administered 240 mg/kg/d of SPG-56, with a significant difference (*p*  <  0.01) compared with the untreated control. The inhibitory effect of SPG-56 against MCF-7 cells was found to be mediated by the altered expression of specific genes. It was concluded that SPG-56 merits further research as a novel anti-tumor agent for breast cancer treatment.

#### 3.4.2. Colon Cancer

Colon cancer is responsible for a high proportion of cancer mortality throughout the world. A new small molecule, glycoprotein SPG-8700, isolated from Zhongshu-1 sweet potatoes [82] was found to promote apoptosis in HCT-116 colon cancer cells by regulating the expression of Bcl-2 and Bax genes, with no effect on normal cell growth. Sporamin, another molecule isolated from sweet potato, has promising effects against colorectal cancer in vitro and in vivo [83]. This proteinase inhibitor was able to modify the gene expression profile of colon cancer cells, up-regulating genes involved in the homeostasis of intracellular metal ions and the activities of essential enzymes and DNA damage repair. 

A study by Lim et al. [84] on the anthocyanin-enriched sweet potato (clone P40) found it offered protection against colorectal cancer by inducing cell cycle arrest, inhibiting proliferation, and apoptotic mechanisms. The anticancer activity of this clone was demonstrated in both in vitro cell culture and an in vivo animal model. Treatment of human colon SW480 cancer cells with a P40 anthocyanin extract resulted in a dose–dependent inhibition of cell proliferation due to a cytostatic but not cytotoxic mechanism.

#### 3.4.3. Other Cancers

Phenolic phytochemicals present in fruits and vegetables indisputably confer anticancer benefits upon regular consumption. The protective effect of PSPA and polyphenol-rich sweet potato extract in other types of cancer has been assessed in studies with cell culture and in vivo.

PSPA has been shown to have in vitro antitumor effects in bladder cancer, a common malignant disease. Li et al. [85] reported that PSPA reduced bladder cancer cell viability in a dose-dependent manner, increasing the apoptosis rate and suppressing the cell cycle. The mechanism of action underlying the anticancer effects of PSPA includes upregulation of pro-apoptosis genes and a lower expression of the anti-apoptotic gene Bcl-2. Thus, the results of this study provide new insights into the treatment of bladder cancer and the potential role that PSPA plays in cancer prevention. 

The effect of PSPA on the bladder cancer cell line BIU87 was also investigated [85]. Compared with the control, the proliferation of BIU87 cells was significantly inhibited in groups treated with this phenolic compound, which induced cell apoptosis in a dose-dependent manner.

The growth-inhibitory and apoptosis-inducing properties of polyphenol-rich sweet potato extract were recently demonstrated in cell culture and in vivo prostate cancer xenograft models [86]. Thus, the extract is a candidate for use as a dietary supplement for prostate cancer management. Despite the growth and apoptosis inhibitory properties of phenolic compounds, future studies should be carried out in humans to support the findings detected in vitro. 

### 3.5. Metabolic Disorders and Intestinal Barrier Function

Anthocyanin and carotenoid-rich extracts from purple and orange-fleshed sweet potatoes, respectively, may be useful as supplementary ingredients for the treatment of obesity and related diseases. Their anti-obesity effects were studied both in vitro (3T3-L1 cells) and in vivo (high-fat diet-induced obese mice) [87]. Both extracts exhibited the potential to inhibit fat accumulation in adipocytes, reduce weight gain, and restore triglyceride levels to normal with an improvement in the ratio between triglyceride and high-density lipoprotein cholesterol, a cardio-metabolic biomarker that predicts a higher risk of heart disease and arteriosclerosis.

Although full evidence from clinical trials is still lacking, and the long-term effects have not been studied yet, sweet potato has proved effective in treating hyperglycemia, as concluded in a recent systematic review based on in vitro and in vivo studies [88]. Cardiovascular disorders are one of the most important causes of mortality in type 2 diabetes mellitus, and atherosclerosis is the major cardiovascular complication related to diabetes [89]. In fact, the endothelial dysfunction found in diabetes has been established as the initial point in the atherosclerotic process [90]. Dietary flavonoids from sweet potatoes were found to attenuate atherosclerosis in mice by alleviating inflammation, inhibiting platelet aggregation, and reducing LDL cholesterol, which improved endothelial function [91]. More data were provided by another study, which reported that flavonoids isolated from purple sweet potato protected against endothelium dysfunction in type 2 diabetes mellitus by reducing endothelial premature senescence and atherogenesis via suppressing ROS levels and the NLRP3 inflammasome [92].

Recently, the sesquiterpene trifostigmanoside I (TS I) was identified as the main compound responsible for the protective effect of sweet potato extracts on intestinal barrier function [5]. Through PKCα/β-ERK1/2 signaling, TS I induced the production of mucin (MUC2), a polypeptide secreted by specialized epithelial cells that forms a protective hydrated gel over mucosal surfaces, and partially protected tight junctions, both effects helping to maintain the intestinal barrier function. More research is needed on other phytochemicals in sweet potato to determine their effect on the regulation of multiple signaling pathways involved in gastrointestinal disorders.

## 4. Effect of Sweet-Potato Processing on Phenolic Compounds 

Sweet potato bioactive compounds have attracted the interest of researchers in the field of human nutrition and the agro-food sector. Unfortunately, the fresh tubers are highly perishable and difficult to conserve due to their high moisture content, sustained metabolism, and microbial attack [93]. The resulting losses generate extra costs for farmers, traders, consumers, and other stakeholders in the sweet potato value chain [18]. Aiming to exploit the economic and nutritional value of this crop more effectively, researchers have joined efforts to explore the impact of different drying and cooking methods on the nutritional composition and physicochemical properties of sweet potatoes. The chemical profile of foods can be affected by several factors, ranging from agronomic practices to processing and storage conditions [94,95]. As phenolic compounds are important constituents of a healthy human diet, their retention in food is desirable from the nutritional point of view. The main findings in terms of phenolic composition are presented in the following sections and in Figure 7.

### 4.1. Drying Treatments

The drying of foods is a hugely important technique for the food industry, as it allows the shelf life of fresh food to be extended and the development of new ingredients and products [96,97]. However, improper drying methods can cause a decline in product quality, including color deterioration, malformation, and loss of nutrients [96].

Convective hot-air and sun-drying are among the methods commonly used to preserve starch-containing products, as they inhibit enzyme activity, prevent microbiological spoilage, and delay decay [93]. However, in addition to being energy- and time-consuming, these processes can also induce loss of nutrients and change the physical and chemical characteristics of the final products [93].

#### 4.1.1. Hot-Air Drying, Microwave Drying, and Vacuum-Freeze Drying

Different drying methods (hot-air, microwave, and vacuum-freeze) were tested in sweet potato tubers to determine their effects on the TPC. In dried sweet potatoes, the TPC was found to be higher than in the fresh tuber, increasing by 116–225%. Microwave-dried samples retained the highest TPC [98], which could be due to the enhanced release of bound phenolics by the heat-induced breakdown of cellular constituents and/or changes in the water content. The increase in phenolic content could also be explained, at least partially, by the formation of Maillard reaction products with a phenolic-type structure during the thermal process [98]. A recent study carried out by Savas [99] compared the effect of freeze-drying and convective drying on the TPC, total flavonoids, and total anthocyanins in sliced sweet potatoes. The TPC and total flavonoid content in freeze-dried samples, and total anthocyanin content in convectively dried samples were clearly higher than in fresh tubers. After the optimization of variables in the convective drying process, this method led to an increase in both types of functional compounds.

#### 4.1.2. Spray Drying

One of the strategies aimed at promoting the consumption of sweet potatoes is the conversion of sweet potato puree into dry powder for use as a functional ingredient in food systems [100]. Freeze-drying and spray drying are commonly used to obtain sweet potato powders. Spray drying, one of the most rapid and least expensive procedures, involves spraying the food material into a chamber where hot dry air rapidly causes the small droplets to evaporate, leaving the spray-dried particles [101]. However, the high temperatures used during the process reduces the (poly)phenol content. In a recent study on purple-fleshed sweet potato, spray drying was found to negatively affect different phenolic groups (anthocyanins and cinnamoylquinic acids) with total (poly)phenol losses representing around 90% of the initial concentrations [102]. The least affected compounds were mono-cinnamoylquinic acids with feruloyl moieties and mono-acylated peonidin derivatives with *p*-hydroxybenzoic acid.

Flour, in addition to being obtained by methods such as hot air-drying, freeze-drying, and microwave-drying, can also be produced by spray-drying. The use of carriers (polysaccharides, proteins, and lipids) was tested as a strategy to improve the properties of spray-dried sweet potato flour [103]. The results showed that apart from increasing flour yields, the use of maltodextrins as a carrier led to a higher retention rate of anthocyanins, flavonoids, and total phenolics compared to flours produced without or with other carriers [103].

### 4.2. Pretreatments

As well as being more stable during storage, dried processed foods can be healthy and nutritious if prepared with the appropriate technologies [104]. Pretreatments before drying and processing can avoid nutrient losses and change the physical and chemical characteristics of the final products, even improving the nutritional value of the sweet potato. Ultrasound (US) [104] and vacuum impregnation (VI) [105,106] are the most frequently applied pretreatments for sweet potatoes.

#### 4.2.1. Ultrasound 

Ultrasound (US) technology is based on acoustic vibration, during which the food undergoes rapid compressions and expansions, the so-called “sponge effect”. This results in alterations in surface tension and viscosity, cell wall disruption, the formation of microscopic channels, and denaturation of enzymes [107,108]. These changes can increase the energy transfer rate in the drying procedure, thereby shortening it, and maintain or improve the product quality [96,109]. 

Despite the importance of phenolic compounds, to date, little research has focused on evaluating the effect of US pretreatment on the phenolic profile of sweet potatoes. One study concluded that pretreatment with US does not alter antioxidant activity (determined by the DPPH method) [110]. The effect of combining US and other pretreatments with drying methods (hot air, microwave vacuum, and freeze) on the stability of quality attributes of sweet potato slices were studied [104]. Compared with the unprocessed samples, the TPC in processed potatoes was found to be lower after 90 days of storage at room temperature. However, it is notable that the reduction in phenolic content was higher in the samples without pre-treatment (76%) than in pretreated (blanching + US in distilled water + hot air drying + microwave vacuum or blanching + US-assisted osmotic dehydration + hot air drying + microwave vacuum) samples (46 and 33%, respectively).

Finally, another study demonstrated that the TPC and total flavonoid content in US-assisted dry sweet potatoes increased compared (15 and 87%, respectively) to control samples when the lowest drying temperature (70 °C) was used [111]. Thus, although US application can accelerate the drying process and maintain the quality of dried products, its positive effect depends on the drying method applied.

#### 4.2.2. Vacuum Impregnation 

Vacuum impregnation (VI) is a non-destructive treatment that takes advantage of the highly porous nature of most foods (i.e., vegetables, fruits, meats, cheese, etc.). Thus, the main purpose of VI is to add substances of interest to porous foods, replacing the native liquids or gases [105,112,113]. Overall, VI may be used to achieve qualitative, technological, or nutritional functionality [105,113]. One of its benefits is that it permits the preparation of a fortified functional food without subjecting the product to a high temperature [113]. VI is used as a pretreatment before drying, freezing, and frying foods to preserve the color, natural flavor, aroma, and heat-sensitive nutrient components [113]. To date, only two studies have evaluated the use of VI in sweet potatoes, with the aim of polyphenol fortification.

Using VI technology (25 min at 20 °C), Abalos et al. (2020) [105] produced sweet potato slices with a 473% increase in phenolic concentration compared to the control. In this study, a commercial solution of polyphenol extract (95% [*v*/*v*] proanthocyanidins) was used as an impregnation medium. The influence of cooking (steam, *cook vide,* and *sous vide*) on the concentration of polyphenols in VI pre-treated samples was also compared, with *sous vide* causing lower losses than cooking under normal atmospheric pressure.

In the second study [106], sweet potatoes were impregnated with freshly squeezed onion juice, kale juice, an onion–kale mixture (1:1), or sodium chloride solution. The samples were then dried using vacuum drying and freeze-drying methods. The pretreatment with VI increased the sweet potato content of polyphenols, carotenoids, chlorophyll, and volatile organic compounds, and improved their antioxidant capacity.

According to the studies carried out so far, VI pretreatment can increase the phenolic content of sweet potatoes, although the final concentration depends on the subsequent method of drying or cooking. 

### 4.3. Cooking Techniques

Sweet potatoes can be cooked by various methods including boiling, steaming, roasting, microwave roasting, and baking. During these processes, the phenolic compounds in sweet potato are susceptible to changes [114], attributed mainly to (i) degradation by heat, (ii) oxidation by polyphenol oxidase, (iii) leaching of water-soluble phenolics, and (iv) isomerization and/or release of compounds by rupture of cell walls [1,19,114,115]. The relation between the type of cooking method and the final phenolic concentration in sweet potatoes has been the subject of various studies [1,19,115,116,117,118,119].

The TPC in sweet potato can be increased by cooking. Roasting, for example, resulted in significantly higher levels compared to the raw vegetable (5.42 mg and 3.34 mg chlorogenic acid equivalent/g DW, respectively) [118]. Other studies have found that steaming, microwaving, baking, and boiling also enhance the TPC [119,120]. After boiling and frying, phenolic compounds were retained in orange-fleshed sweet potatoes [18,121], whereas steam-cooking drastically increased the TPC in purple sweet potatoes (about four-fold) compared to raw tubers [115]. This could be due to the softening or disruption of plant cell walls or the breakdown of complex compounds during thermal treatment, which facilitates the release of phenolic compounds [118,120]. 

Other studies have compared the effect of different cooking methods without including a control (raw sweet potato). In a comparison [1] of domestic processing techniques (baking, boiling, frying, microwaving, sautéing, and steaming), the TPC was least affected by boiling, whereas the biggest reductions were induced by deep frying, followed by microwaving. In purple-fleshed sweet potato, baking and steaming did not influence the TPC, but boiling resulted in a significant decrease [117]. Conversely, in orange-fleshed sweet potato, boiling, and roasting had a less negative effect on the TPC than steaming [19]. 

A clear trend has not been observed in the effect of cooking on phenolic acids in sweet potato, which include chlorogenic acids, a family of esters formed from certain cinnamic acids and quinic acid. A study by Carrera et al. (2021) demonstrated that orange-fleshed sweet potatoes had higher concentrations of caffeoylquinic acids (4-caffeoylquinic acid and 4,5-dicaffeoylquinic acid) after oven-roasting compared with boiling or steaming in the microwave [116]. The fact that boiled sweet potatoes generally contain a lower concentration of caffeoylquinic acids may be due to the extraction effect of water during cooking or by temperature. The same study demonstrated that phenolic compounds decreased with prolonged cooking in water. Another comparative study reported that total caffeoylquinic acids were reduced by different home processing techniques in the following order: boiling > deep-frying > sautéing or steaming > microwaving or oven baking [1]. Finally, a positive effect of cooking on the chlorogenic acid content was detected in a purple-fleshed variety; the biggest increase was found in sweet potatoes cooked by steaming, followed by baking > microwaving > boiling [120].

Anthocyanins are well known to be sensitive to heat and light and susceptible to variations in pH and the presence of oxygen, but the way they are affected by cooking depends on the technique used [114,122]. The highest loss of anthocyanins in purple-fleshed sweet potato was reported to be induced by baking, followed by steaming and boiling [117]. In contrast, another study found that boiling, steaming, microwaving, and baking increased the content in almost all tested samples (orange and purple) compared to raw sweet potato [119]. An increase in total flavonoids was also found in boiled and fried, orange-fleshed sweet potatoes [18]. Finally, steam-cooking of purple sweet potato had a positive effect on total monomeric anthocyanin and flavonoid contents, which increased by approximately 13-fold and 5-fold, respectively, compared to the control [115]. 

Regarding the carotenoids, food processing can change their levels in this vegetable. Abong et al. (2021) reported that after frying and boiling sweet potatoes retained more than 90% of their *β*-carotene content [18]. Additionally, boiling has been found to reduce *trans-*β-carotene but increase *cis-*β-carotene [114]. 

In summary, the effect of cooking on the TPC, individual phenolic compounds, and carotenoids in sweet potato depends on the technique. The temperature and time of cooking and contact with water should be controlled to minimize the loss of these bioactive compounds.

## 5. Conclusions

*Ipomoea batata* L.; Lam. is an important food crop with a comprehensive content of macronutrients and micronutrients, and it stands out as a dietary source of several active secondary metabolites, principally carotenoids, phenolic acids, tocopherols, anthocyanins, flavonoids, and coumarins. Overall, it can be concluded that the concentration of phytochemicals in sweet potato depends on the variety, as well as the processing and storage conditions. 

Growing evidence for the protective health effects of its bioactive constituents has attracted new interest in this root vegetable from the food industry, consumers, and scientists. The powerful antioxidant capacity of purple sweet potatoes is mainly attributed to their anthocyanin and carotenoid content. These compounds have a high capacity to eliminate free radicals and contribute to the inhibition of lipid peroxidation activity, offering protection against injuries induced by hepatotoxins. In addition, consumption of sweet potatoes rich in anthocyanins is closely associated with a lower risk of diabetes, cardiovascular disease, and cancer, and has also been associated with improved cognitive function. Some minor sesquiterpene compounds of sweet potato have also been described to improve intestinal barrier function. Although the protection of the purple sweet potato anthocyanin from injury induced by hepatotoxins has been studied in healthy men with borderline hepatitis, most of the studies describing the potential of the sweet potato phytochemicals have been performed in vitro and in animal models, so human intervention trials are necessary to demonstrate the possible health effect in humans. Therefore, further research is needed in this field, especially in randomized clinical trials in humans. The challenge today is to evaluate the effect of prolonged consumption of different sweet potatoes, especially purple ones, on indicators of diseases such as oxidative stress. Positive results would open future market prospects and encourage consumers to adopt sweet potatoes as a functional food with the ability to prevent the incidence of chronic diseases in the long term.

Finally, the application of commercial processing (drying) and domestic cooking methods (boiling, frying, steaming, baking, and microwaving) can affect the functional properties of sweet potatoes to different degrees. Although studies have shown that the concentration of phenolic compounds can change during processing (due to oxidation, leaching, isomerization, degradation, and/or release from cell walls), no clear trend has been observed as to what constitutes the optimal strategy for their preservation or enhancement. Therefore, further research is required on the effects of processing and cooking on sweet potatoes to provide consumers with guidelines on how to maximize their health properties.

## Figures and Tables

**Figure 1 antioxidants-11-01648-f001:**
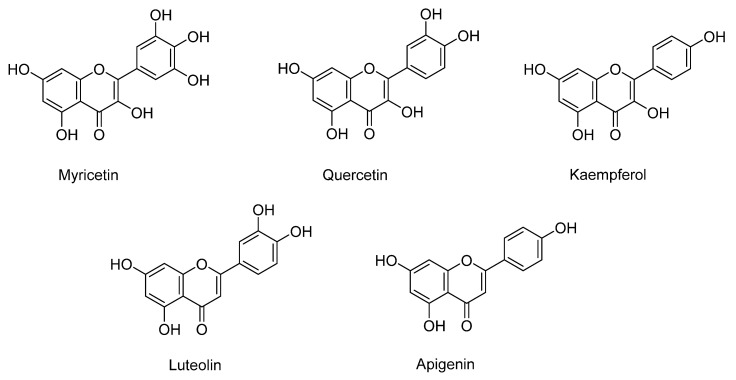
The main flavonoids identified in sweet potato.

**Figure 2 antioxidants-11-01648-f002:**
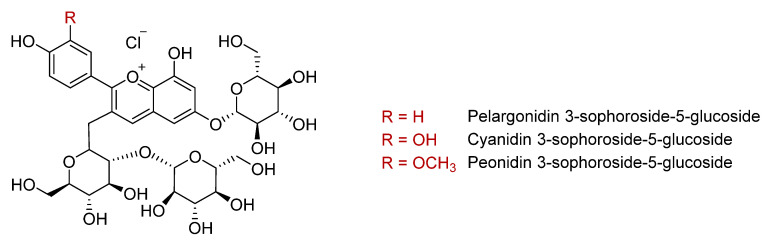
Main anthocyanins in purple sweet potato.

**Figure 3 antioxidants-11-01648-f003:**
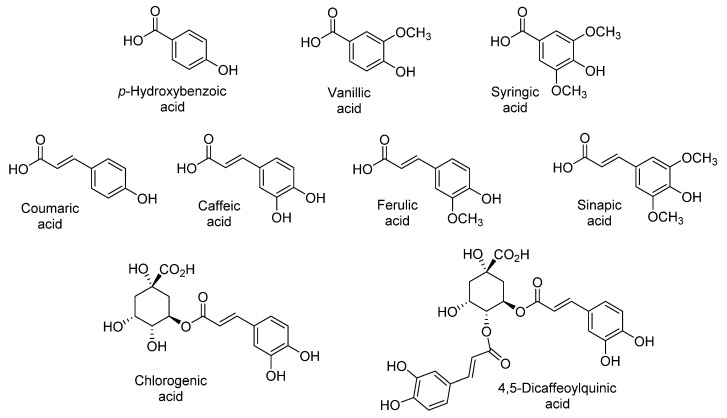
Main phenolic acids identified in sweet potato.

**Figure 4 antioxidants-11-01648-f004:**
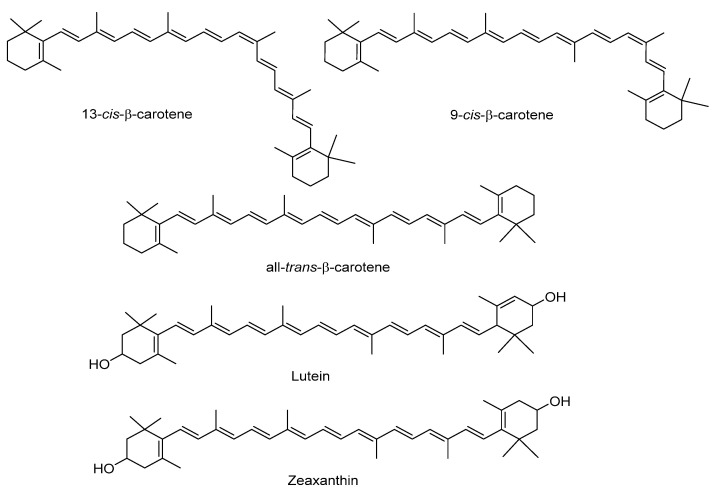
Main carotenoids identified in sweet potato.

**Figure 5 antioxidants-11-01648-f005:**
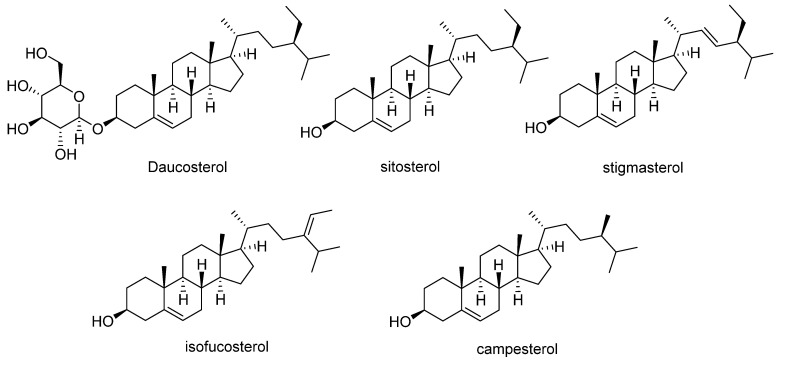
Main phytosterols identified in sweet potato.

**Figure 6 antioxidants-11-01648-f006:**
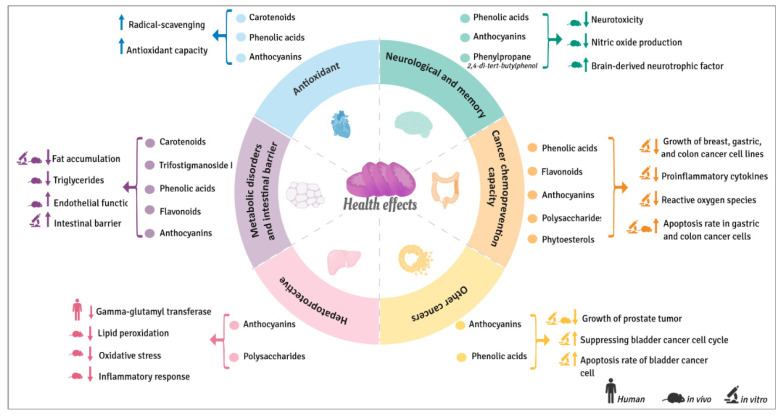
Beneficial health effects of bioactive compounds from sweet potatoes.

**Figure 7 antioxidants-11-01648-f007:**
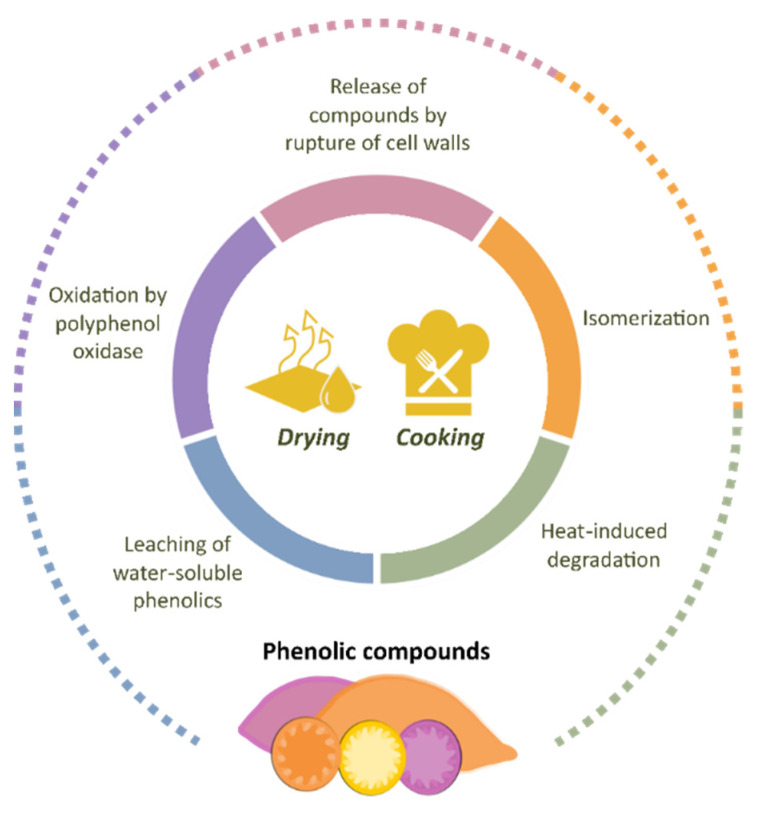
Main factors affecting the phenolic profile in sweet potato during processing.

**Table 2 antioxidants-11-01648-t002:** (Poly)phenol content of purple sweet potato flesh.

Origin	Sample Extraction	Analytical Method	Phytochemical	Amount of Phytochemical	Ref.
USA	Methanol (80%)	Folin-Ciocalteu	TPC	~0.2 to 0.7 mg CA/g FW	[9]
		pH-differential	TCA	~0.1 to 0.4 mg TCA/g FW	
Korea	0.2% HCl in methanol	UHPLC-(ESI)-Qtof, UPLC-Ion trap, and HPLC -DAD	Peo-3-O-glc	6544 to 26,483 mg/kg DW	[22]
			Cya -3-O-glc	943 to 3962 mg/kg DW	
			Pg-3-O-glc	1242 to 2181 mg/kg DW	
	0.2% HCl in methanol	UHPLC-(ESI)-QqQ	Phenolic acids: caffeic acid, ferulic acid, chlorogenic acid, *p*-coumaric acid, *cis*-ferulic acid, *trans*-ferulic acid, caffeoylquinic acid, and dicaffeoylquinic acids.	Phenolic acids (mg/Kg DW): Caffeic acid: 44 to 70, *cis*-ferulic acid; 2 to 7, *trans*-ferulic acid: 1 to 7, chlorogenic acid: 6714 to 13268, *p*-coumaric acid: 1, caffeoylquinic acid: 5150 to 5862, dicaffeoylquinic acids 19 to 24.	[22]
			Flavonoids: quercetin 3-O-galactoside, and quercetin-3-O-glc, quercetin diglc.	Flavonoids (mg/kg DW): Quercetin 3-O-galactoside:1, quercetin-3-O-glc: 1 to 7, quercetin diglc: 6 to 30.	
	0.2% HCl in methanol	Folin–Ciocalteu	TPC	1.80 to 7.37 mg GA/100 g DW	[22]
Japan	Ethanol (80%)	Folin–Ciocalteu	TPC	0.2 to 1.2 µmol CA/mL.	[28]
		Reverse-phase HPLC	Peo and Cya	NS	
China	Water, 3.5% citric acid, and 79 U/mL cellulose	HPLC- MS/MS	Cya-based anthocyanins and peo-based anthocyanins	13.7 mg total anthocyanins /100 g	[29]
USA	7% acetic acid in methanol (80%)	Folin–Ciocalteu	TPC	408.1 mg CA/100 g FW (raw)	[30]
				401.6 mg CA/100 g FW (puree)	
	7% acetic acid in methanol (80%)	pH-differential	Total monomeric anthocyanin	101.5 mg cya-3-glc/100 g FW (raw) 80.2 mg cya-3-glc/100 g FW (puree)	[30]
China	Methanol (85%) with 0.5% formic acid	LC−PDA−APCI−MS	Total monomeric anthocyanins: cya 3-soph-5-glc, cya 3-(6′′-*p*-caffeoylsoph)-5-glc, peo 3-soph-5-glc, cya 3-(6′′-*p*-feruloylsoph)-5-glc, peo 3-(6′′-*p*-feruloylsoph)-5-glc	305.0 mg anthocyanins/100 g DW	[31]
			Hydroxycinnamic acid derivatives: caffeoyl-hexoside, 5-*O*-caffeoylquinic acid, caffeic acid, feruloylquinic acid, 3,4-di-*O*-caffeoylquinic acid, 3,5-di-*O*-caffeoylquinic acid	854.4 mg hydroxycinnamic acids/100 g DW	
Korea	Methanol (50%) with 1.2 M HCl at 80 °C	HPLC system	Flavonoids: quercetin, myricetin, kaempferol, luteolin.	Flavonoids: 579.5 µg/g DW (Quercetin: 388.9, myricetin: 152, kaempferol: 23.4, luteolin: 15.2)	[27]
			Anthocyanins: Cya, Peo.	Anthocyanins: 727.4 µg/g DW (Cya: 408.4, and Peo: 319.1)	
			Phenolic acids: ferulic, *p*-coumaric, *p*-hydroxybenzoic, sinapic, syringic, and vanillic acids.	Phenolic acids: 744.3 µg/g DW (*p*-hydroxybenzoic acid: 238.6, vanillic acid: 147.4, syringic acid: 3.9, *p*-coumaric: 18.1, ferulic acid: 322.3, sinapic acid: 14.1)	
India	Different extraction solvents: methanol/trifluoroacetic acid (TFA) (99.5:0.5), ethanol/TFA (99.5:0.5), methanol/TFA/water (80:19.5:0.5), and ethanol/TFA/water (80:19.5:0.5)	HR-ESI–MS	TAC	43.4 mg peonidin-3-*O*-glc equivalent /100 g of FW	[32]
Japan	Methanol/acetic acid (19:1, *v*/*v*), methanol/water (1:1, *v*/*v*), and *tert*-butyl methyl ether/methanol (7:2, *v*/*v*)	HPLC-DAD, HPLC-ESI-MS^n^	Cya 3-soph-5-glc (Cya-3-(6′′-caffeoylsoph)-5-glc, cya-3-(6′′-caffeoylsoph)-5-glc, cya-3-(6′′-caffeoyl-6′′′-feruloylsoph)-5-glc, cya-3-feruloylsoph-5-glc)	NS	[33]
			Peo3-soph-5-glc (Peo-3-(6′′-caffeoylsoph)-5-glc, peo-3-feruloylsoph-5-glc, peo-3-(6′′,6′′′-dicaffeoylsoph)-5-glc, peo-3-(6′′-caffeoyl-6′′′-feruloylsoph)-5-glc, peo-3-(6′′-caffeoyl-6′′′-*p*-hydroxybenzoylsoph)-5-glc, peo-3-*p*-hydroxybenzoylsoph-5-glc)		
China	Ethanol with 1% formic acid	UPLC-PDA, UPLC-QTOF-MS, UPLC-MS/MS analyses	TAC	90.5 to 1018 mg/100 g DW	[31]
			Monoacylated anthocyanin	0.0 to 44.8 mg/100 g DW	
			Diacylated anthocyanin	79.9 to 982.9 mg/100 g DW	
			Acylated-based anthocyanin	90.5 to 1018.7 mg/100 g DW	
			Cya-based anthocyanin	25.7 to 326.6 mg/100 g DW	
			Peo-based anthocyanin	0.0 to 761.7 mg/100 g DW	
Korea	Methanol with 0.2% HCl	HPLC-TOF/MS, HPLC/MS/MS, and UV/vis spectroscopy	TAC	383.2 to 1190.2 mg/100 gDW	[22]
			Non-acylated anthocyanin	17.5 to 35.8 mg/100 gDW	
			Monoacylated anthocyanin	158.2 to 323.4 mg/100 gDW	
			Diacylated-based anthocyanin	199.6 to 845.1 mg/100 gDW	
			Cya-based anthocyanin	98.2 to 815.1 mg/100 gDW	
			Peo-based anthocyanin	281.5 to 740.8 mg/100 gDW	
			Pg-based anthocyanin	1.2 to 217.0 mg/100 gDW	
Korea	5% formic acid in water	LC-DAD-ESI/MS	TAC	Raw: 1342 mg/100 g DW	[34]
				Steamed 751 mg/100 g DW	
				Roasted 1086 mg/100 g DW	
USA	5% formic acid water	HPLC/MS-MS	TAC	Raw: 1390 mg/100 g DW	[35]
				Baked: 1303 mg/100 g DW	
				Steamed: 1284 mg/100 g DW	
				Microwaved: 1275 mg/100 g DW	
				Pressured cook: 1165 mg/100 g DW	
				Fried: 1217 mg/100 g DW	
			Total Cya content (Cya 3-*p*-hydroxybenzoyl soph -5-glc, cya 3-(6″-caffeoyl soph)-5-glc, cya 3-(6″ -feruloyl soph)-5-glc, cya 3-(6″,6″′-dicaffeoyl soph)-5-glc, cya 3-caffeoyl-*p*-hydroxybenzoyl soph -5-glc, cya 3-(6″-caffeoyl-6″′-feruloyl soph)-5-glc)	Raw: 930 mg/100 g DW	
				Baked: 943 mg/100 g DW	
				Steamed: 1060 mg/100 g DW	
				Microwaved: 1038 mg/100 g DW	
				Pressured cook: 943 mg/100 g DW	
				Fried: 937 mg/100 g DW	
			Total peo content (Peo 3-*p*-hydroxybenzoyl soph-5-glc, peo 3-(6″-feruloyl soph)-5-glc, peo 3-caffeoyl soph -5-glc, peo 3-caffeoyl-*p*-hydroxybenzoyl soph -5-glc, peo 3-(6″-caffeoyl-6″′-feruloyl soph)-5-glc)	Raw: 460 mg/100 g DW	
				Baked: 360 mg/100 g DW	
				Steamed: 224 mg/100 g DW	
				Microwaved: 237 mg/100 g DW	
				Pressured cook: 222 mg/100 g DW	
				Fried: 280 mg /100 g DW	
China	Methanol:Water (7:3, *v*/*v*)	HPLC-MS	5-caffeoylquinic acid, 3,5-dicaffeoylquinic acid, 4,5-dicaffeoylquinic acid	NS	[36]

CA: chlorogenic acid; Cya: cyanidin; DE: dry extract; DW: dry weight; ESI: electrospray ionization; FW: Fresh weight; Glc: glucoside; HPLC: high-performance liquid chromatography; MS: mass spectrometry; NS: non-specified; PDA: photodiode array detection; Peo: peodinin Pg: Pelargonidin; QTOF: quadrupole-time-of-flight; Soph: sophoroside; UPLC: ultra-performance liquid chromatography; TPC: Total phenolic content; TFC: Total flavonoid content; TAC: Total anthocyanin content.

**Table 3 antioxidants-11-01648-t003:** (Poly)phenol content of white, yellow, and red sweet potato flesh.

Sweet Potato Flesh Color (Origin)	Sample Extraction	Analytical Method	Phytochemical	Amount of Phytochemical	Ref.
White (USA)	Methanol (80%)	Folin-Ciocalteu	TPC	<0.1 mg CA/g FW	[9]
White and orange (Italy)	Methanol	Folin-Ciocalteu	TPC	Raw: 794 mg GA/kg DW	[24]
				Boiled: 1803 mg GA/kg DW	
				Fried: 2605 mg GA/kg DW	
				Microwaved: 1836 mg GA/kg DW	
				Steamed: 1743 mg GA/kg DW	
White-fleshed (Korea)	50% MeOH withn1.2 M HCl at 80 °C	HPLC system	Flavonoids: quercetin, myricetin, kaempferol.	Flavonoids: 45.4 µg/g DW (Quercetin: 19.8, myricetin: 23.4, kaempferol: 2.1)	[27]
			Phenolic acids: ferulic, *p*-coumaric, *p*-hydroxybenzoic, sinapic, syringic, and vanillic acids.	Phenolic acids: 52.5 µg/g DW (*p*-hydroxybenzoic acid: 5.5, vanillic acid: 7.5, syringic acid: 3.7, *p*-coumaric: 7.5, ferulic acid: 15.1, sinapic acid: 13.3)	
Yellow (USA)	Methanol (80%)	Folin-Ciocalteu	TPC	<0.1 mg CA/g FW	[9]
Red (Peru)	Methanol	Folin-Ciocalteu	TPC	945 mg CA/100 g FW 3220 mg CA/100 g DW	[21]
	0.225 N HCl in ethanol (95%)	pH-differential	Total monomeric anthocyanins (Cya 3-glc)	182 mg total anthocyanins/g FW 618 mg total anthocyanins/g DW	
Red	NS	pH-differential	TAC	2.4 to 40.3 mg total anthocyanins/g FW	[37]

CA: chlorogenic acid; Cya: cyanidin; DW: dry weight; Fresh weight; GA: gallic acid; Glc: glucoside; HPLC: high-performance liquid chromatography; NS: non-specified; TPC Total phenolic content; TFC: Total flavonoid content; TAC: Total anthocyanin content.

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
