# Peer review of "Sweet Potato Is Not Simply an Abundant Food Crop: A Comprehensive Review of Its Phytochemical Constituents, Biological Activities, and the Effects of Processing†"

_antioxidants, 2022, doi:10.3390/antiox11091648_

Round 1
Reviewer 1 Report
The authors have proposed a comprehensive review on sweet potato phytochemical constituents, biological activities, and the effects of processing. The work is technical sound and the authors utilized appropriate techniques for the selection of the articles and the authors are experts in the field. Moreover the review can be of great interest for a the readers of the journal. Overall it is interesting and easy to follow.
However, following suggestions are recommended:
-there are several typewriting errors and some sentences are rambling. Please correct.
The tables (1 and 2) are two big. Please break down in to three or four tables, otherwise they are very difficult to follow.
- In the last paragraph of the introduction, the Author needs to more clearly state the novelty of this review with regards to the others present in literature.
-the conclusion section must be improved to better explain the potentiality of the phytochemicals described in the above sections and their applications together with future prospects.
Author Response
Reviewer 1
The authors have proposed a comprehensive review on sweet potato phytochemical constituents, biological activities, and the effects of processing. The work is technical sound and the authors utilized appropriate techniques for the selection of the articles and the authors are experts in the field. Moreover the review can be of great interest for a the readers of the journal. Overall it is interesting and easy to follow.
However, following suggestions are recommended:
-there are several typewriting errors and some sentences are rambling. Please correct.
Thanks for the suggestion. We have performed changes to correct some typewriting mistakes. For example, “Pakistan” instead of “Pakinstan”; “affected” instead of “effected”; “acylated anthocyanin” instead of “acylatedanthocyanin”; “cis” instead of “Cis”. Additionally, we have incorporated some letters in italics, and corrected some were/was mistakes. We have also unified the style of the abbreviations in table footnotes and performed some little changes to better understand the text.
The tables (1 and 2) are two big. Please break down in to three or four tables, otherwise they are very difficult to follow.
Thanks for the suggestion. To make it easier for the reader to find information, we have divided the first table into 3 tables and the second into 2 tables according to the color of the sweet potato flesh. Thus, the manuscript now has 5 tables:
- Table 1. (Poly)phenol content of orange sweet potato flesh
- Table 2. (Poly)phenol content of purple sweet potato flesh
- Table 3. (Poly)phenol content of white, yellow, and red sweet potato flesh
- Table 4. Carotenoid, phytosterol, and other phytochemical contents of orange sweet potato flesh.
- Table 5. Carotenoid content of purple, yellow, and white sweet potato flesh.
- In the last paragraph of the introduction, the Author needs to more clearly state the novelty of this review with regards to the others present in literature.
Following the suggestion, we have changed the last paragraph of the introduction by this text and we have cited a new reference: “Recent reviews on sweet potato leaves covered current knowledge of their bioactive composition and possible health effects [1] and the protocols developed for the plant regeneration as an alternative method to produce disease-free planting material [Planta, 2022, 256(2), 40. https://doi.org/10.1007/s00425-022-03938-8 ]. Beyond the widely studied potential of sweet potato leaves and their cultivars practices, sweet potato roots are more than just foods used for centuries as a major source of carbohydrates. Today they are recognized as highly nutritious and useful food for the prevention of chronic diseases. To corroborate this, we have integrated for the first-time updated information on their content in bioactive compounds, and an extensive revision of the in vitro and in vivo evidence of its benefits for human health. In addition, since phenolic compounds are important constituents of a healthy human diet, we have explored the effect of sweet potato processing on phenolic compounds.“
-the conclusion section must be improved to better explain the potentiality of the phytochemicals described in the above sections and their applications together with future prospects.
Following the reviewer's suggestion, we have changed the text of the conclusions to clarify how the phytochemicals in sweet potatoes are related to their potential health benefits and comment better on the future prospects in this field. This is the new text:
Growing evidence for the protective health effects of its bioactive constituents has attracted new interest in this root vegetable from the food industry, consumers, and scientists. The powerful antioxidant capacity of purple sweet potatoes is mainly attributed to their anthocyanin and carotenoid content. These compounds have a high capacity to eliminate free radicals and contribute to the inhibition of lipid peroxidation activity, offering protection against injuries induced by hepatotoxins. In addition, consumption of sweet potatoes rich in anthocyanins is closely associated with a lower risk of diabetes, cardiovascular disease, and cancer, and has also been associated with improved cognitive function. Some minor sesquiterpene compounds of sweet potato have also been described to improve intestinal barrier function. Although the protection of the purple sweet potato anthocyanin from injury induced by hepatotoxins has been studied in healthy men with borderline hepatitis, most of the studies describing the potential of the sweet potato phytochemicals have been performed in vitro and in animal models, so human intervention trials are necessary to demonstrate the possible health effect in humans. Therefore, further research is needed in this field, especially in randomized clinical trials in humans. The challenge today is to evaluate the effect of prolonged consumption of different sweet potatoes, especially purple ones, on indicators of diseases such as oxidative stress. Positive results would open future market prospects and encourage consumers to adopt sweet potatoes as a functional food with the ability to prevent the incidence of chronic diseases in the long term.
Reviewer 2 Report
This review summarizes the most recently published information focusing on bioactive phytochemical constituents, potential effects on health, and the impact of processing technologies of the sweet potatoes. The manuscript reviews 122 articles regarding this topic. The topic of this manuscript is up to date, attractive and well suited for your journal. The manuscript is well written and divided into 5 main parts, the text is clear and easy to read. For better visualisation authors used 7 figures and 2 complex tables. I suggest checking for some spelling mistakes and grammar errors. Otherwise, I have no major concerns about this manuscript and I recommend it for publication.
Author Response
Reviewer 2
This review summarizes the most recently published information focusing on bioactive phytochemical constituents, potential effects on health, and the impact of processing technologies of the sweet potatoes. The manuscript reviews 122 articles regarding this topic. The topic of this manuscript is up to date, attractive and well suited for your journal. The manuscript is well written and divided into 5 main parts, the text is clear and easy to read. For better visualisation authors used 7 figures and 2 complex tables.
I suggest checking for some spelling mistakes and grammar errors. Otherwise, I have no major concerns about this manuscript and I recommend it for publication.
Thanks for the suggestion. We have performed changes to correct some typewriting mistakes. For example, “Pakistan” instead of “Pakinstan”; “affected” instead of “effected”; “acylated anthocyanin” instead of “acylatedanthocyanin”; “cis” instead of “Cis”. Additionally, we have incorporated some letters in italics, and corrected some were/was mistakes. We have also unified the style of the abbreviations in table footnotes and performed some little changes to better understand the text.